# The Interfacial Adhesion Performance and Mechanism of a Modified Asphalt–Steel Slag Aggregate

**DOI:** 10.3390/ma13051180

**Published:** 2020-03-06

**Authors:** Wenhuan Liu, Hui Li, Huimei Zhu, Pinjing Xu

**Affiliations:** College of Materials Science and Engineering, Xi’an University of Architecture and Technology, Xi’an 710055, China; zhuhuimeitj@163.com (H.Z.); xpj6100@163.com (P.X.)

**Keywords:** asphalt–steel slag interface, steel slag aggregate, adhesion, physical adsorption, chemical reaction

## Abstract

The interfacial adhesion between asphalt and steel slag aggregate is a decisive factor in the formation of an asphalt–steel slag mixture and significantly affects the quality stability of steel slag–asphalt mixtures. In this study, the adhesion between an asphalt and steel slag aggregate, the interfacial microstructure, the adsorption and desorption characteristics, and chemical reactions were, respectively, explored by a PosiTestAT–A adhesion puller, a scanning electron microscope, a net adsorption test, an infrared spectrometer, and a dynamic shear rheometer. The mechanism of adhesion between the asphalt and steel slag aggregate was analyzed from the perspectives of physical adsorption and chemical reactions. The results showed that different factors had different effects on the adhesion of asphalt–steel slag aggregate interface. The freeze–thaw cycle and steel slag aggregate particle size had significant effects on interfacial adhesion, while the asphalt heating temperature, water bath time, and stirring time had relatively weak effects on interfacial adhesion. Compared to a limestone aggregate, the steel slag–asphalt mixture had greater adhesion and better adhesion performance because the pits and textures on the surface of the steel slag aggregate produced a skeleton–like effect that strengthened the phase strength of the asphalt–slag aggregate interface, thereby improving the adhesion and increasing the physical adsorption between the asphalt and steel slag aggregate. In addition, due to the N–H stretching vibrations of the amines and amides, as well as SiO–H stretching vibrations, a chemical reaction occurred between the asphalt and steel slag aggregate, thus improving the adhesion performance between the asphalt and steel slag. Based on the shape of the adsorption isotherm, it was determined that the adsorption type was multi–molecular layer adsorption, indicating that the adhesion between the asphalt and steel slag mainly involved physical adsorption.

## 1. Introduction

The interactions between asphalt and the steel slag aggregate are important in the formation of an asphalt mixture’s structure and are directly related to the strength, temperature stability, water stability, and aging speed of an asphalt mixture [1,2,3,4,5]. The interfacial strength between the asphalt and steel slag significantly affects the quality stability of steel slag–asphalt mixtures [6,7,8,9]. The phase structure, interaction, and adhesion properties of asphalt mixtures have been extensively explored using various methods such as kinetics, thermodynamics, and surface physical chemistry [10,11,12,13]. The adhesion of asphalt mixtures increases with an increase in asphalt grade. A small quantity of mineral powder can improve the adhesion of an asphalt mortar, whereas a large quantity of mineral powder can adsorb light oil components in the asphalt, thus reducing adhesion. The interactions between the asphalt and the aggregate are mainly physical interactions. Using an adhesion test, Podoll [14] proved that the peeling of asphalt occurs at the aggregate interface. Ling [15] carried out an adhesion test with asphalt and stone and analyzed the variations of adhesion with shear rate and temperature tests. Tarrer [16] indicated that when steel slag comes into contact with asphalt, the former has good adhesion characteristics since steel slag is an alkaline aggregate, and asphalt contains acidic groups. By exploring the asphalt–filler adhesion characteristics by infrared spectroscopy, Shin [17] found that asphalt around the filler carries polar functional groups. Xu [18] studied the combustion characteristics of asphalt by combining thermogravimetry with infrared spectroscopy.

Other researchers have indicated that the angularity coefficient and fractal dimension of the surface texture of a steel slag aggregate have a linear positive correlation with the phase strength of the asphalt–aggregate interface. The greater the surface roughness of the steel slag aggregate, the higher the interfacial strength. Kong [19] studied asphalt–aggregate interface characteristics from the three perspectives of aggregate chemical composition, aggregate surface texture characteristics, and aggregate surface free energy characteristics. Using a scanning electron microscope, Kim [20] analyzed the fracturing and self–healing processes of an asphalt mixture. Blair [21] analyzed porous media with a scanning electron microscope and determined the porosity and specific surface area of the material. Williams [22] analyzed the peeling of asphalt from its aggregate surface in an asphalt mixture with a scanning electron microscope after a freeze–thaw cycle test. Tasong [23] explored the effect of the aggregate surface structure on the phase structure of an asphalt–aggregate interface by SEM and XRD, and found that the surface texture, roughness, and shape of the mineral powder affected the interface morphology of the asphalt cement.

Although several studies on asphalt mixtures have been carried out, quantitative studies on the adhesion at the asphalt–slag aggregate interface, the formation mechanism of the interface structure, and the adsorption and desorption characteristics of the asphalt–slag aggregate interface are rare. Therefore, in this study, the adhesion between the asphalt and steel slag aggregate, the interface microstructure, the adsorption and desorption characteristics, and chemical actions were, respectively, explored using a PosiTestAT–A adhesion puller, a scanning electron microscope, a net adsorption test, and an infrared spectrometer [24,25,26].

## 2. Experiment

### 2.1. Materials and Reagents

The materials and reagents used in this study included converter steel slag powder (Shaanxi Longmen Iron and Steel Co., Ltd., Xi’an, China), A–70 road petroleum asphalt (Hebei Xingtai Hansheng Asphalt Sales Co., Ltd., Xingtai, China), limestone (Jiangxi Kete Fine Powder Co., Ltd., Jiangxi, China), 42.5R Portland Cement (Shaanxi Ecological Cement Co., Ltd., Xi’an, China), standard sand (Xiamen Aiso standard sand Co., Ltd., Xiamen, China), and toluene solution (Shanghai Yiyan Biotechnology Co., Ltd., Shanghai, China).

The steel slag aggregate used in the test was obtained by grinding with a vertical high–pressure roller mill. Its chemical composition, physical and mechanical properties, and XRD diffraction patterns are, respectively, shown in Table 1, Table 2, and Figure 1.

The high–viscosity modified asphalt used in this test was prepared with A–70 road petroleum asphalt and a high–viscosity modifier. As shown in Table 3, all the indicators of this asphalt met the requirements of the Technical Specifications for Permeable Asphalt Pavement (CJJT 190–2012).

### 2.2. Experimental Methods

The adhesion test was performed according to the following method. First, steel slag and limestone aggregate were ground into a flat surface by a bench–type grinder polishing machine and then cured with cement mortar. Quartz sand was used, and the cement mortar was prepared according to the following mixing ratio (water:cement:coarse sand:medium sand:fine sand): 2:4:3:4.5:4.5. The steel slag and limestone aggregate particles were pressed into cement mortar in such a way that the ground sides of the particles were positioned upward. The aggregate surface was slightly higher than the mortar surface. After 7 days of curing at 20 °C and an RH of 95%, a thin layer of asphalt was applied to the upper surface of the steel slag, and the adhesion between the asphalt and steel slag was then determined with a PosiTestAT–A adhesion puller.

The morphology of the asphalt–slag interface in the asphalt mixture was observed using a Quanta200 scanning electron microscope produced by the American FEI Company (Hillsboro, OR, USA).

A 721N visible light spectrometer was used to test the adsorption and desorption characteristics of the asphalt–slag interface under the visible light wavelength of 370 nm with a quartz cuvette at 25 ± 1 °C. In the experiment, 500 mL of toluene solution was added into a 1000 mL conical flask. The asphalt was weighed for the test and stirred using a heat–collecting, constant–temperature–heating magnetic stirrer until the asphalt was completely dissolved. A quartz cuvette filled with pure toluene solution was put into a visible light spectrophotometer to adjust the light transmittance to 100% and the absorbance to zero. The prepared asphalt–toluene solution was stirred, and 4 mL of the prepared solution was then transferred to a 25 mL test tube. After adding 16 mL of pure toluene, the test tube was shaken well and put into a visible spectrophotometer to measure its absorbance. Subsequently, 50 g of dried steel slag aggregate was added into a conical flask and stirred at a temperature of 25 °C. Every 1 h, 4 mL of the mixture solution was sampled, transferred into a 25 mL test tube, and diluted with a certain quantity of toluene to obtain 0.1 g/L asphalt to determine the absorbance.

An FT–IR infrared spectrometer was used to explore the interactions between the asphalt and steel slag. The infrared spectra were collected in the range of 400~4000 cm^−1^ with a resolution of 2 cm^−1^. The steel slag–asphalt mixture was prepared according to the following procedure. First, 1202.29 g of steel slag particle aggregate with a grading specification of 4.75~9.5 mm and 105 g of steel slag powder filler were mixed, dried to a constant weight in an oven at 170 °C, and poured into an E10-H asphalt mixer. Next, 4.19 g of a high-modulus anti-rutting agent was added and uniformly mixed for 180 s. Next, 87.91 g of the petroleum asphalt was added and uniformly mixed for 180 s.

DSR (Dynamic Shear Rheological) test: The change rule of the phase angle of the mixture containing modified asphalt and steel slag mineral components was analyzed using a dynamic shear rheometer. The strain of the dynamic shear rheometer was 12%, the scanning frequency was 0.1 Hz~10 Hz, the gap width of the DSR sample was 1.00 ± 0.05 mm, and the diameter of the DSR sample was 25 mm; the test temperatures were 45 °C and 55 °C. The modified asphalt sample was heated to 140 °C and kept at a constant temperature for about 1 h. The steel slag mineral phase C_2_S (dicalcium silicate), C_3_S (calcium silicate), and RO powder samples (the RO phase is a broad solid solution formed by melting FeO, MgO, and other divalent metal oxides, such as MnO) were dried to a constant weight at 105 °C and kept at 150 °C for 30 min. The asphalt and C_2_S, C_3_S, and RO phase powders were mixed (the ratio between the asphalt and mineral powder was 2.5:1) with a low-speed mixer at 140 °C for 10 min until bubbles no longer appeared on the surface of the mixture, thus ensuring uniform mixing of the asphalt and mineral phase.

### 2.3. Performance Test and Characterization

The mechanical properties of the steel slag and asphalt were tested according to the Technical Specifications for Permeable Asphalt Pavement (CJJT 190–2012) [27]. The chemical composition was determined with an S4 PIONEER X-ray fluorescence analyzer (Bruker Corporation, Karlsruhe City, Baden-Württemberg, Germany). The mineral composition was determined with a D–MAX/2500 X-ray diffractometer (Rigaku Corporation, Zhaodao City, Tokyo, Japan). The adhesion of the asphalt–steel slag interface was measured with a PosiTestAT–A puller (DeFelsko Corporation, New York City, NY, USA). The adsorption and desorption characteristics of the asphalt–steel slag interface were determined with a 721N visible spectrometer (Tianjin Terus Technology Co., Ltd. Tianjin, China). The reactions between the asphalt and steel slag were tested with a Fourier infrared spectrometer (PERKINELMER Corporation, Waltham City, MA, USA), and the microscopic morphology was observed with a Quanta 2000B scanning electron microscope (FEI Corporation, Hillsboro City, OR, USA). The change rule of the phase angle of the mixture of modified asphalt and steel slag mineral components was analyzed using an AR–2000 DSR (TA Instruments Corporation, New Castle DE City, DE, USA).

## 3. Results and Discussion

### 3.1. Adhesion Analysis of Steel Slag–Asphalt Interface

Figure 2 shows the influence of the particle diameter of the steel slag on adhesion at the asphalt–slag interface. When the particle diameter decreased from 25~40 mm to 10~15 mm, the adhesion decreased by about 3.6% from 12.54 MPa to 12.10 MPa, which was still higher than the adhesion between the limestone and asphalt, indicating that the increased particle diameter of the steel slag led to increased adhesion between the steel slag and asphalt. This adhesion effect was better than that of the limestone aggregate. Figure 3 shows the effect of asphalt temperature on the adhesion at the asphalt–slag interface. When the asphalt stirring temperature increased from 160 °C to 180 °C, the adhesion between the steel slag and asphalt first increased and then decreased. When the asphalt temperature was 170 °C, the adhesion between the steel slag and asphalt was the greatest (12.51 MPa). When the asphalt temperature was 160 °C, the adhesion between the steel slag and asphalt was the weakest (12.33 MPa). This result showed that the asphalt heating temperature had a weak effect on the adhesion between the steel slag and asphalt.

Figure 4 shows the effects of mixing time on the adhesion at the asphalt–slag interface. When mixing time was increased from 30 min to 90 min, the adhesion between the steel slag aggregate and asphalt gradually increased by about 1.1% from 12.32 MPa to 12.46 MPa, indicating that the adhesion was increased by increasing mixing time. However, the trend of this increase was not obvious. Figure 5 shows the effect of a water bath environment on the adhesion at the asphalt–slag interface. When the water bath time was increased from 30 min to 48 h, the adhesion gradually decreased by about 1.3% from 12.28 MPa to 12.12 MPa, indicating that increasing water bath time reduced adhesion, but this reduction was not obvious.

Figure 6 shows the effect of the freeze–thaw cycles on the adhesion at the asphalt–slag interface. When the number of freeze–thaw cycles was increased from one to five, the adhesion decreased rapidly by about 11.90% from 12.22 MPa to 10.92 MPa, indicating that the adhesion was decreased by increasing the freeze–thaw cycles. With an increase in the cycles, the decreasing trend of the adhesion was enhanced. Figure 7 shows the effect of the addition of steel slag powder on the adhesion of the asphalt–slag interface. When the addition proportion of fine steel slag powder was increased from 0% to 50%, the adhesion gradually increased by about 2% from 11.28 MPa to 11.51 MPa. Therefore, the adhesion was increased by increasing the addition of fine steel slag powder.

### 3.2. Microscopic Morphology of Asphalt–Aggregate Interface

Figure 8 shows the micro-morphology of the interfaces between the asphalt and steel slag and between the asphalt and limestone. Tiny pits and cracks on the steel slag surface were filled with asphalt to increase the embedding depth on the steel slag surface (Figure 8a). The limestone surface was dense, and the asphalt was not embedded into the limestone surface (Figure 8b). Due to the difference in the surface microstructures of the steel slag and limestone, the structure of the asphalt–steel slag interface was different from that of the asphalt–limestone interface. As a result, the contact conditions between the asphalt and steel slag were different than those between asphalt and limestone. After the asphalt was embedded in the aggregate surface to a certain depth, its stress area increased, greatly enhancing its resistance to external forces. Therefore, the embedded asphalt on the aggregate surface enhanced the aggregate’s interfacial strength. The physical structural characteristics of the steel slag surface provided a skeleton-like effect for the asphalt–aggregate interface. A stronger skeleton effect corresponded to a higher interface strength and a greater interface bonding force between the steel slag and asphalt.

The steel slag and limestone aggregates were studied via the mercury intrusion method. The particle diameter of the aggregate was controlled to about 10 mm. The results are shown in Table 4. The steel slag had a larger porosity and specific surface area, as well as a more uniform pore size distribution, than the limestone. A thicker and more uniform asphalt film was thus adsorbed onto the surface of the steel slag, thereby improving adhesion between the steel slag and asphalt film and enhancing the protection of the asphalt film on the aggregate when an external load was applied to the asphalt mixture.

### 3.3. Adsorption and Desorption Characteristics between Asphalt and Aggregate

Figure 9 shows the relationship between the net asphalt adsorption capacity and the adsorption time. Under the same testing conditions, the net asphalt adsorption capacity of the steel slag showed the same trend as that of the limestone, except that the capacity of the former was greater than the latter. Some small pits and pores on the steel slag surface enhanced the adsorption effect.

At a constant temperature and a certain concentration of adsorbate, the adsorption capacity on the solid surface is fixed. The adsorption isotherm can be obtained by measuring the corresponding adsorption capacity under a series of relative concentrations. The adsorption isotherm reflects the adsorption characteristics and pore structure on the solid surface and can be used to explore the properties of the surface pores and the type of adsorption. Figure 10 shows the variation in net asphalt adsorption capacity with adsorption time under different equilibrium concentrations. Figure 11 shows the asphalt–slag aggregate adsorption isotherms. With an increase in the equilibrium concentration, the asphalt adsorption capacity gradually increased and remained basically stable after 4 h of adsorption. During the initial 2 h of desorption, water stripped asphalt from the steel slag surface, and the amount of desorption gradually increased. However, due to the moisture evaporating in the mixed solution, the net adsorption capacity increased with an increase in desorption time. The asphalt–slag adsorption time was 4 h, and the desorption time was 2 h, indicating that the adsorption involved multi-layer adsorption.

### 3.4. Analysis of Reaction Characteristics between Modified Asphalt and Steel Slag

Figure 12 shows the FTIR spectra of the asphalt–steel slag mixture. As can be seen, the asphalt had N–H or O–H absorption characteristic peaks at 3200 cm^−1^ to 3700 cm^−1^, typical carboxylic acid absorption characteristic peaks at 2500 to 3200 cm^−1^, alkane and cycloalkane C–H absorption characteristic peaks at 2920 cm^−1^ and 2850 cm^−1^, and methyl –CH_3_ absorption characteristic peaks at 2950 cm^−1^. The absorption characteristic peak of methylene –CH_2_– appeared at 2920 cm^−1^ and 2850 cm^−1^, the absorption characteristic peak of N–H appeared at 1560 cm^−1^ to 1640 cm^−1^, the absorption characteristic peak of the C–CH_3_ asymmetric bond and –CH_2_– symmetric bond appeared at 1460 cm^−1^ and 1375 cm^−1^, and the absorption characteristic peak of amide C–N appeared at 1400 to 1420 cm^−1^. The absorption characteristic peak of aliphatic amine C–N appeared at 1030~1280 cm^−1^, that of the aromatic ether compound Ar–O appeared at 1260 cm^−1^, that of the sulfoxide S=O functional group appeared at 1030 cm^−1^, that of N–H appeared at 650~900 cm^−1^, and aromatic absorption characteristic peaks appeared at 800 cm^−1^ and 860 cm^−1^. This shows that the asphalt contained alkane, cycloalkane, carboxylic acid, ester, amide, aliphatic amine, aromatic ether, sulfoxide, and other functional compounds, of which some (such as carboxyl) were hydrophilic functional groups. The steel slag had absorption characteristic peaks of N–H or O–H at 3200–3700 cm^−1^, –CH_3_ at 1300–1600 cm^−1^, C–H, Si–O, and Si–H at 750–1200 cm^−1^, C–H, N–O, and C–S at 750–500 cm^−1^, and S–S at 400–500 cm^−1^. The absorption characteristic peak of active hydrogen appeared at 3407 cm^−1^, the absorption characteristic peak of –CH_2_ appeared at 1430cm^−1^, and the absorption characteristic peak of alkane –CH(CH_3_)_2_ appeared at 921 cm^−1^. The absorption characteristic peak of the meta-disubstituted aromatic hydrocarbon C–H appeared at 710 cm^−1^, and the absorption characteristic peak of C–N–O appeared at 500 cm^−1^, which indicated that the steel slag contained functional compounds such as hydrocarbons, aromatics, silica, sulfur, and oxygen.

Furthermore, the broad and weak absorption peaks of the modified asphalt–steel slag mixture between 3200 cm^−1^ and 3700 cm^−1^ were ascribed to N–H or O–H bonds disappearing and the intensity of the absorption peaks at 2500 cm^−1^ to 3200 cm^−1^ decreasing, indicating that the intermolecular interactions of the modified asphalt components were weakened. A new absorption peak between 3500 cm^−1^ and 3750 cm^−1^ was caused by the N–H stretching vibrations of amines and SiO–H. This new peak indicated that the interaction between the asphalt and the steel slag powder was also caused by a certain chemical reaction. It can also be concluded from Figure 12 that, although there was a chemical reaction between the asphalt and steel slag interface, most absorption peaks indicated the superposition of the absorption peaks of the asphalt and steel slag, and no obvious new stretching vibration peaks or deformation vibration peaks were observable, proving that physical action was the main action between the asphalt and steel slag aggregate.

Figure 13 shows the phase angles of the modified asphalt and C_2_S, C_3_S, and RO phases detected by DSR at different temperatures. It can be seen from Figure 13a that the phase angle of the mixture after adding the C_2_S, C_3_S, and RO phases was not significantly reduced compared to that of the asphalt sample at 45 °C. This showed that at 45 °C, the chemical reaction between the C_2_S, C_3_S, RO phase, and asphalt sample was not significant. It can be seen from Figure 13b that the phase angle of the mixture after adding the C_2_S and RO phase was not significantly lower than that of the asphalt sample at 55 °C, while the phase angle of the mixture after adding C_3_S was significantly lower. This showed that at 55 °C, the chemical reaction between the C_3_S and the asphalt sample was obvious, but the chemical reaction between the C_2_S, the RO phase, and the asphalt sample was not obvious. As can be seen from Figure 13, the phase angle variation range of the different asphalt and mineral phases between 45 °C and 55 °C was as follows: the phase angle of the asphalt sample was 76.39°~87.75°; the phase angle of the C_2_S–asphalt was 75.75°~88.05°; the phase angle between the RO phase and the asphalt was 76.30°~88.08°; and the phase angle of the C_3_S–asphalt was 70.61°~86.95°. After adding C_3_S, the phase angle changed most significantly, followed by the addition of C_2_S, and, finally, the RO phase. This showed that the chemical reaction between the C_3_S and the asphalt was obvious, but the chemical reaction between the C_2_S, the RO phase, and the asphalt sample was not obvious.

## 4. Conclusions

The strength of the asphalt–steel slag interface is a key factor affecting the stability of a steel slag–asphalt mixture. In this study, the adhesion properties of the steel slag–asphalt interface were explored. Based on the results above, the following conclusions were drawn.

Compared with the limestone–asphalt mixture, the steel slag–asphalt mixture showed superior adhesion performance. The adhesion was greatest when the asphalt heating temperature was 170 °C. With an increase in freeze–thaw cycles, the decreasing trend of adhesion was gradually enhanced. Mixing time had a weak effect on the adhesion at the asphalt–slag interface.

Unlike the microstructure of the asphalt–limestone interface, the steel slag surface contained rich tiny pits, cracks, and porous structures, which provided a larger infiltration interface for the asphalt mortar and facilitated a greater embedding and anchoring depth on the steel slag surface. In this way, the asphalt film thickness and the structural asphalt ratio were increased, and a solid steel slag–asphalt interface structure was formed. Consequently, the adhesion between the asphalt and steel slag aggregate was enhanced.

The asphalt adsorption effect of the steel slag aggregate was significantly higher than that of the limestone aggregate. The adsorption process involved multi-molecular layer adsorption, indicating that the adhesion between the asphalt and steel slag aggregate mainly entailed physical adsorption.

Due to amine and amide N–H stretching vibrations and SiO–H stretching vibrations, chemical reactions occurred between the asphalt and steel slag aggregate, thereby improving the adhesion performance between the asphalt and steel slag aggregate. The chemical reaction between the C_3_S and asphalt was obvious, and the chemical reactions between the C_2_S, the RO phase, and the asphalt sample was not obvious.

## Figures and Tables

**Figure 1 materials-13-01180-f001:**
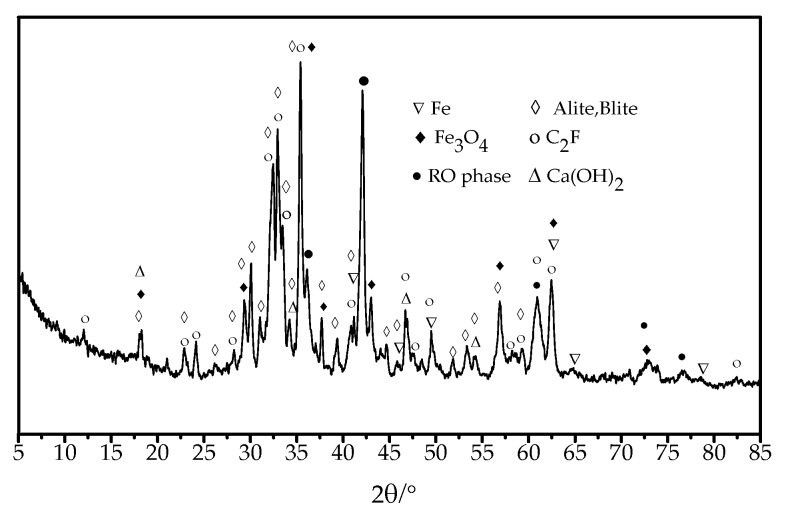
XRD test results for the steel slag aggregate.

**Figure 2 materials-13-01180-f002:**
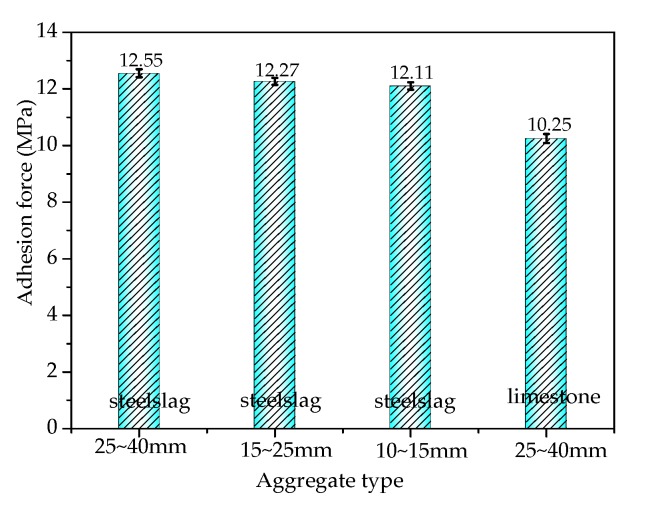
Effect of the particle diameter of steel slag on the adhesion at the asphalt–steel slag interface.

**Figure 3 materials-13-01180-f003:**
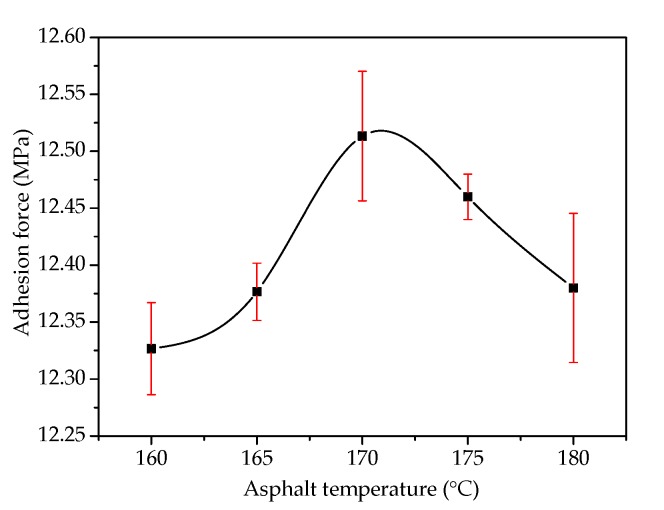
Effect of asphalt temperature on the adhesion at the asphalt–steel slag interface.

**Figure 4 materials-13-01180-f004:**
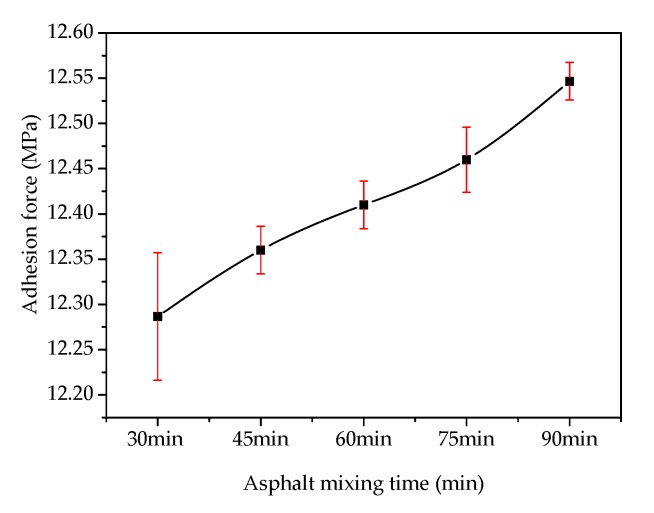
Effect of mixing time on the adhesion at the asphalt–steel slag interface.

**Figure 5 materials-13-01180-f005:**
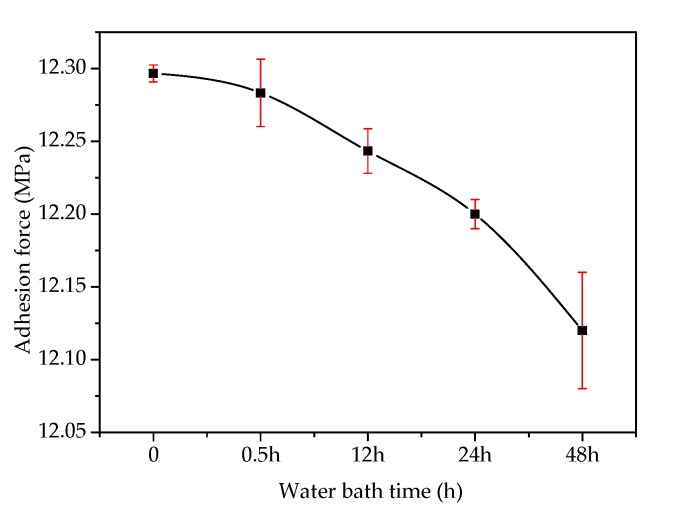
Effect of the water bath environment on the adhesion at the asphalt–steel slag interface.

**Figure 6 materials-13-01180-f006:**
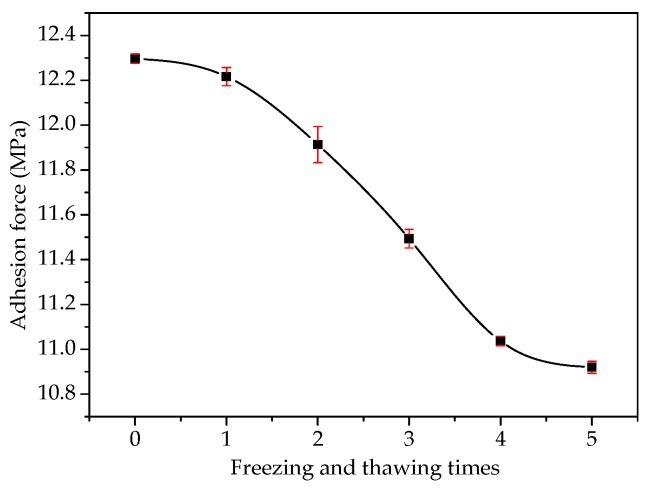
Effect of freeze–thaw cycles on the adhesion at the asphalt–steel slag interface.

**Figure 7 materials-13-01180-f007:**
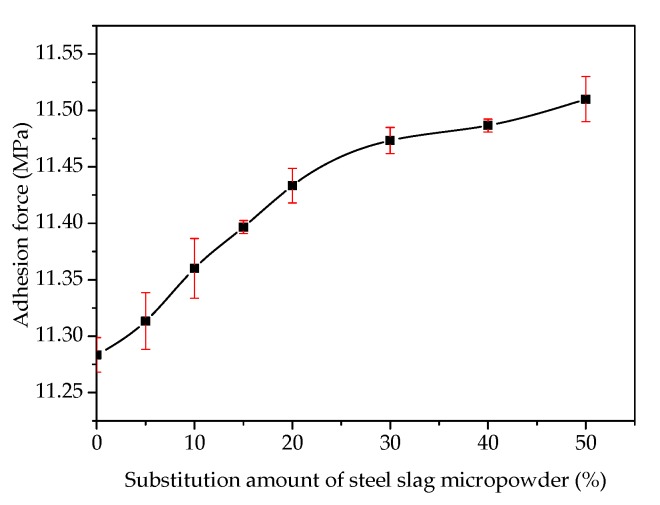
Effect of the addition of steel slag powder on the adhesion at the asphalt–steel slag interface.

**Figure 8 materials-13-01180-f008:**
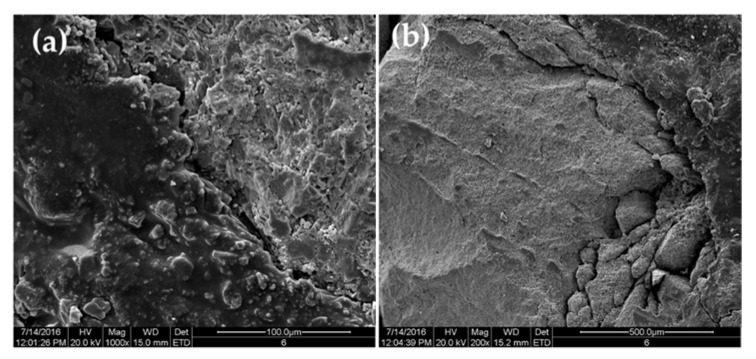
SEM images of asphalt–aggregate interfaces: (**a**) asphalt–steel slag interface and (**b**) asphalt–limestone interface.

**Figure 9 materials-13-01180-f009:**
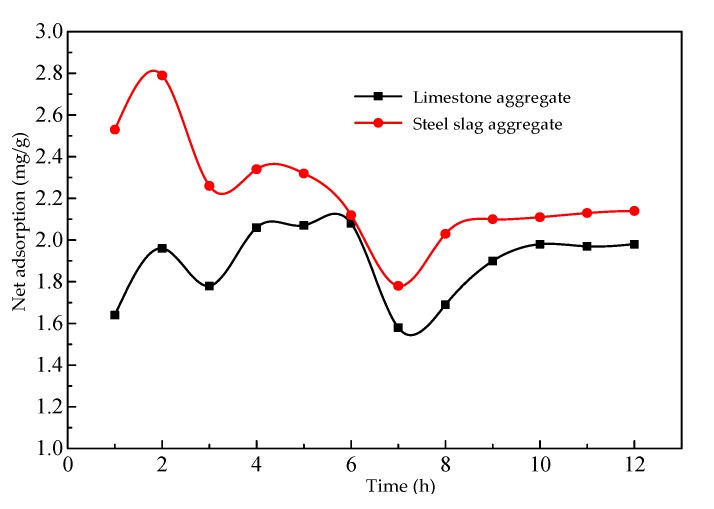
Relationship between net asphalt adsorption capacity and adsorption time.

**Figure 10 materials-13-01180-f010:**
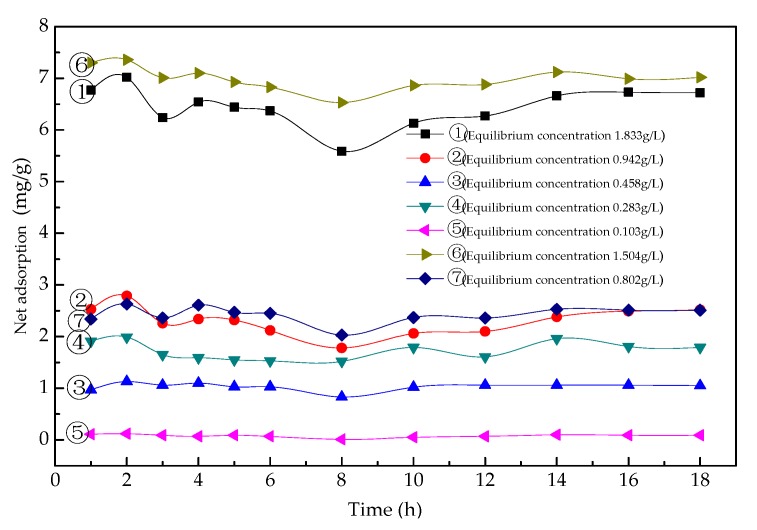
Relationship between the net asphalt adsorption capacity of the steel slag and time under different equilibrium concentrations.

**Figure 11 materials-13-01180-f011:**
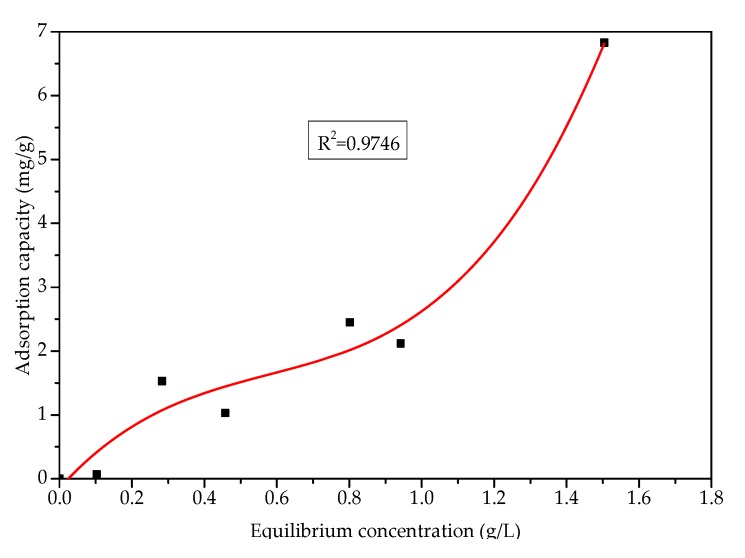
Adsorption isotherms of the asphalt–steel slag.

**Figure 12 materials-13-01180-f012:**
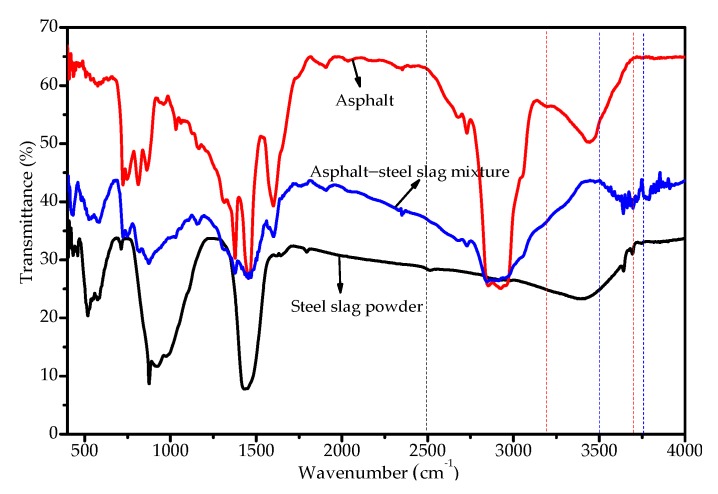
FT–IR spectra of the asphalt–steel slag mixture.

**Figure 13 materials-13-01180-f013:**
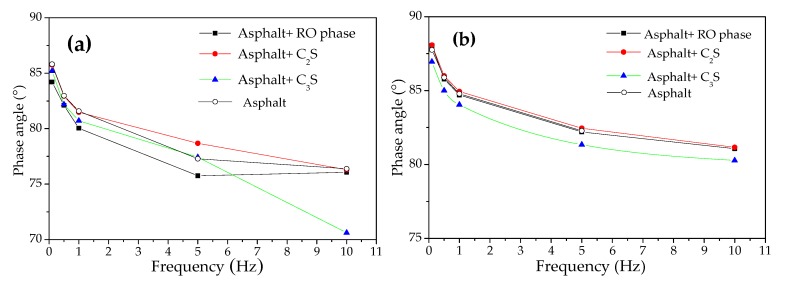
Phase angles of the asphalt and mixtures of C_2_S, C_3_S, and RO at different temperatures. (**a**) The temperature was 45 °C. (**b**) The temperature was 55 °C.

**Table 1 materials-13-01180-t001:** XRF test results for the steel slag (wt.%).

CaO	Fe_2_O_3_	SiO_2_	Al_2_O_3_	MgO	MnO	P_2_O_5_	TiO_2_	SO_3_	Na_2_O	K_2_O	Others
44.83	21.65	14.38	5.48	3.42	1.94	0.83	0.57	0.23	0.05	0.04	6.58

**Table 2 materials-13-01180-t002:** Mechanical properties of the steel slag and limestone.

Test Item	Steel Slag	Limestone	Request	Normative References of the Tests
Apparent relative density (g/cm^3^)	3.39	2.93	≥2.90	CJJT 190–2012
Water absorption (%)	1.83	0.80	≤3.0	CJJT 190–2012
Needle particle content (%)	4.56	3.62	≤12	CJJT 190–2012
Aggregate crushing value (%)	13.9	13.9	≤26	CJJT 190–2012
Water washing method <0.075 mm (%)	0.2	0.1	≤1.0	CJJT 190–2012
Los Angeles abrasion loss (%)	13.2	16.2	≤26	CJJT 190–2012
Incorruptibility (%)	2.6	1.2	≤12	CJJT 190–2012
Soaking expansion rate (%)	1.2	0.9	≤2.0	CJJT 190–2012
Adhesion to asphalt (%)	5	5	≥4	CJJT 190–2012
f–CaO (%)	1.7	/	≤3.0	CJJT 190–2012

**Table 3 materials-13-01180-t003:** Main technical indicators of the asphalt.

Test Item	Measured Value	Request	Normative References of the Tests
Penetration (25 °C, 100 g, 5 s) (0.1 mm)	50.2	≥40	CJJT 190–2012
Ductility (5 cm/min, 10 °C) (cm)	42	≥30	CJJT 190–2012
softening point	85	≥80	CJJT 190–2012
Flash point (COC) (°C)	275	≥260	CJJT 190–2012
60 °C dynamic viscosity (Pa·s)	32,000	≥20,000	CJJT 190–2012

**Table 4 materials-13-01180-t004:** Mercury intrusion porosimetry test of steel slag and limestone.

Test Item		Steel Slag	Limestone
Porosity (by volume) (%)		14.5413	10.6205
BET surface area (m^2^·g^−1^)		4.181	0.193
Proportion of apertures of different size (μm)	>10	19.784	99.273
	1~10	15.479	0.023
	0.1~1	19.397	0
	0.01~0.1	36.570	0.203
	0.001~0.01	8.770	0.501
Medium pore diameter (μm)		0.0138	0.0071
Average pore diameter (μm)		0.0396	1.1821

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
