# Peer review of "The Interfacial Adhesion Performance and Mechanism of a Modified Asphalt–Steel Slag Aggregate"

_materials, 2020, doi:10.3390/ma13051180_

Round 1

Reviewer 1 Report

Dear authors,

congratulation to this interesting study. I have only some small requests/proposals to add additional information regadring the experimental methodology:

  • 2.2, please give some additional details about the procedure of PosiTestAT-A:
    • In which way, the tension force is applied (desctribe test apparatus)?  
    • What is the size of the pulling device (adesion area), which may be important regarding the surface area of the tested aggregates
    • Was the surface area ratio (aggregate / cement mortar) assessed, which may explain the effect of aggregate size
    • A figure could be helpful
  • 2.2, DSR tests:
    • Please give the ratio between bitumen and mineral powder
    • Please add the DSR sample dimension applied (gap width, diameter)

Regarding the results, I did not understand table 5. Perhaps it could be deleted, an the relevant and discussed wavenumbers could be added as vertical lines to figure 12. 

Reviewer 2 Report

materials-738521

Please find some comments on the submitted manuscript below.

The abstract seems more like conclusions chapter. Should have more introductory information and research significance (one sentence for each) and less detailed information on the findings.

In introduction chapter, relevant outcomes of cited literature for the presented study are generally missing. If the aim is to prove the lack of studies in the addressed research field, maybe it is preferable to write something like “Although several studies on asphalt mixes have been carried out, as those by […],  the quantitative studies on the adhesion…

Ln 92 “were” instead of “was”

Ln 108-9 check sentence for grammar

Ln 139 “were” instead of “was”

Subchapter 2.3 seems a repetition of information and therefore disposable.

Fig.s 1-7 force unit is not MPa, but Newton.

 Ln 179-83 It would be interesting to know from slag and limestone, which would perform better concerning freeze-thaw

Ln 180-90 check sentence meaning

Ln 203 check sentence meaning

Author Response

Dear professor, here is my reply to your question and I hope it can meet your requirements.

 1.The abstract seems more like conclusions chapter. Should have more introductory information and research significance (one sentence for each) and less detailed information on the findings.

Answer: According to your suggestion, the author has revised the abstract of the manuscript and put more emphasis on the introductory information and research significance.

2.In introduction chapter, relevant outcomes of cited literature for the presented study are generally missing. If the aim is to prove the lack of studies in the addressed research field, maybe it is preferable to write something like “Although several studies on asphalt mixes have been carried out, as those by […], the quantitative studies on the adhesion…

Answer: This section has been amended in the article.

3.Ln 92 “were” instead of “was”; Ln 108-9 check sentence for grammar; Ln 139 “were” instead of “was”

Answer: All have been modified in the article.

4.Subchapter 2.3 seems a repetition of information and therefore disposable.

Answer: Subchapter 2.3does not repeat

5.Fig.s 1-7 force unit is not MPa, but Newton.

Answer: Fig.s 1-7 force unit is Mpa. In this paper, PosiTest AT-A instrument is used for pull-off test. The numerical unit of liquid crystal digital display of this instrument has been unified into MPa or Psi. The technical parameters of PosiTestAT-A adhesion tester are shown in table 1.

Table1. The technical parameters of PosiTestAT-A

Spindle size

10mm

14mm

20mm*

50mm

Measuring range

0~70MPa

0~40MPa

0~20MPa

0~3.5MPa

0~10000psi

0~6000psi

0~3000psi

0~500psi

Resolution

±0.01MPa(1psi)

Accuracy

±1% Full Range

6.Ln 179-83 It would be interesting to know from slag and limestone, which would perform better concerning freeze-thaw

Answer: This is an illustration of the asphalt-steel slag interface, not slag and limestone.

7.Ln 180-90 check sentence meaning; Ln 203 check sentence meaning

Answer: All have been modified in the article.